# Aβ-Induced Damage Memory in hCMEC/D3 Cells Mediated by Sirtuin-1

**DOI:** 10.3390/ijms21218226

**Published:** 2020-11-03

**Authors:** Haochen Liu, Yixuan Zhang, Hong Zhang, Sheng Xu, Huimin Zhao, Xiaoquan Liu

**Affiliations:** Center of Drug Metabolism and Pharmacokinetics, China Pharmaceutical University, Nanjing 210009, China; haochenliu@cpu.edu.cn (H.L.); yxzhang@stu.cpu.edu.cn (Y.Z.); Hong_Zhang_199x@126.com (H.Z.); sheng_xu_199x@126.com (S.X.); huimin_zhao_199x@126.com (H.Z.)

**Keywords:** cerebrovascular endothelial damage memory, sirt-1, vicious circle, kinetics process modeling, Alzheimer’s disease

## Abstract

It is well accepted by the scientific community that the accumulation of beta-amyloid (Aβ) may be involved in endothelial dysfunction during Alzheimer’s disease (AD) progression; however, anti-Aβ anti-bodies, which remove Aβ plaques, do not improve cerebrovascular function in AD animal models. The reasons for these paradoxical results require investigation. We hypothesized that Aβ exposure may cause persistent damage to cerebral endothelial cells even after Aβ is removed (referred to as cerebrovascular endothelial damage memory). In this study, we aimed to investigate whether cerebrovascular endothelial damage memory exists in endothelial cells. hCMEC/D3 cells were treated with Aβ_1–42_ for 12 h and then Aβ_1–42_ was withdrawn for another 12 h incubation to investigate whether cerebrovascular endothelial damage memory exists in endothelial cells. A mechanism-based kinetics progression model was developed to investigate the dynamic characters of the cerebrovascular endothelial damage. After Aβ_1–42_ was removed, the sirt-1 levels returned to normal but the cell vitality did not improve, which suggests that cerebrovascular endothelial damage memory may exist in endothelial cells. Sirt-1 activator SRT2104 and NAD^+^ (Nicotinamide Adenine Dinucleotide) supplement may dose-dependently relieve the cerebrovascular endothelial damage memory. sirt-1 inhibitor EX527 may exacerbate the cerebrovascular endothelial damage memory. Kinetics analysis suggested that sirt-1 is involved in initiating the cerebrovascular endothelial damage memory; otherwise, NAD^+^ exhaustion plays a vital role in maintaining the cerebrovascular endothelial damage memory. This study provides a novel feature of cerebrovascular endothelial damage induced by Aβ.

## 1. Introduction

Dementia is considered one of the biggest threats to the aging population and a major public health problem worldwide, whose leading cause, Alzheimer’s disease (AD), accounts for about 80% of dementia cases [1]. Previous research has proposed an amyloid cascade hypothesis, which suggested that amyloid β (Aβ) plays a central role in the development of AD [2,3]. To test this hypothesis, multiple anti-bodies that target Aβ (e.g., Solanezumab, Bapineuzumab, and Crenezumab) were tested in AD patients [4]. Unfortunately, none of these anti-bodies exhibited efficacy in clinical trials [5].

An often-cited explanation for the failure of anti-bodies targeted at Aβ in clinical trials is that they are used too late in the disease process [5,6]; however, alternative explanations for clinical failure exist. It is widely accepted that the cerebrovascular system may play an important role in Aβ clearance [7]. Qi et al. suggested that the failure of anti-Aβ immunotherapies may be due to cerebrovascular damage, which cannot be improved by removing Aβ [8]. Previous reports could provide more evidence for this hypothesis. When AD animal models were treated with anti-Aβ antibodies, they were found to be effective at removing Aβ plaques, but not at preventing hemorrhages that may be related to cerebrovascular damage [9,10,11,12]. In other words, Aβ may impair the cerebrovascular function, but the cerebrovascular function cannot be improved by removing Aβ [10]. To provide new insight into the failure of anti-Aβ immunotherapies, it is necessary to investigate the reasons for the lack of efficacy in removing Aβ on cerebrovascular function improvement.

Diabetes “metabolic memory” phenomenon may provide useful information for the investigation of the persistent endothelial dysfunction of AD. The metabolic memory phenomenon is defined as the persistence of diabetes complications even after glycemic control has been pharmacologically achieved [13]. The metabolic memory phenomenon is associated with endothelial dysfunction [14]. In other words, the endothelial dysfunction induced by hyperglycemia in the early stage of diabetes might not be improved by glycemic control. Endothelial dysfunction, which cannot be improved by removing Aβ, seems to function similarly to the metabolic memory phenomenon of diabetes; therefore, it is reasonable to assume that the damage memory phenomenon may exist in cerebrovascular endothelial cells.

Previous research has emphasized the important roles of epigenetic factors in AD [15]. For example, a lot of clinical research has suggested that the DNA methylation levels of some genes could be potential biomarkers in AD. A range of studies has indicated that histone modifications play a vital role in the development of AD. Especially, histone deacetylases (HDACs) were found to have a significant influence on memory formation and cognition [15]; therefore, it is reasonable to assume that the epigenetic factors may be involved in the formation of cerebrovascular endothelial damage memory. Epigenetic factors include DNA methylation, histone modifications, chromatin remodeling, and regulation by non-coding RNA [15]. Among these factors, histone modifications variations are observed in a wide range of research involving AD patients, AD animal models, and AD culture models, which suggests that histone modifications may play a vital role in the development of AD [15,16]. There are multiple types of histone modifications e.g., acetylation, methylation, phosphorylation, and ubiquitination, among which acetylation is the most ubiquitous and well-studied [15,16]. Histone acetylation is catalyzed by histone acetyltransferase (HAT), while deacetylation is influenced by histone deacetylases [15]. Among these HDACs enzymes, sirt-1, which is found to decrease significantly in AD patients, is closely associated with the proliferation and apoptosis of endothelial cells [17,18]; therefore, sirt-1 may be related to the formation of AD cerebrovascular endothelial damage memory. Furthermore, we assumed that sirt-1 may be involved in the formation of endothelial damage via the mitochondria. Decreased Sirt-1 activity may increase acetylated histone H3 binding to the p66^SHC^ promoter and induce overexpression of p66^SHC^. The increased p66^SHC^ would increase the reactive oxygen species (ROS) level and open the mitochondrial permeability transition pore (PTP), which may result in the collapse of the mitochondrial membrane potential (MMP). When the PTP opens, the contact between the cytosolic and the mitochondrial pools of pyridine nucleotides may reduce NAD+ via enzymatic reactions, which may further impair the activity of Sirt-1 and initiate the vicious circle of damage memory.

In this study, we aimed to investigate whether the damage memory process exists in cerebrovascular endothelial cells and to understand the kinetics character of this process. This study contains four steps (Figure 1). First, cell experiments were performed to investigate whether the damage memory exists in endothelial cells and to obtain the data for the kinetics process of the damage of cerebrovascular endothelial cells. Second, a mathematical model was developed to describe the above kinetics process. Third, simulations based on the above model were performed to investigate the kinetics character of the damage process and improvement method of cerebrovascular endothelial cells damage. Fourth, the improvement method proposed by the above simulations was validated by cell experiments. Our research provides new insight into the AD cerebrovascular endothelial cell dysfunction and improvement of cerebrovascular endothelial function.

## 2. Results

### 2.1. Withdrawing Aβ Does Not Improve hCMEC/D3 Cell Vitality

In this study, hCMEC/D3, a well-established in vitro cerebral endothelial model, was selected to investigate that whether Aβ may induce the endothelial damage memory. The results (Figure 2A) show that the cell vitality (measured by MTT (Methyl Thiazolyl Tetrazolium) assay) in the Aβ group decreases during Aβ_1–42_ incubation. After Aβ_1–42_ is withdrawn, the cell vitality in the damage memory group did not recover and there was no significant difference (*p* > 0.05, *t* = 0.28, *n* = 6, *t*-test) compared with the Aβ group. These results suggest that the damage memory may exist in endothelial cells. In other words, if an endothelial cell is exposed to Aβ_1–42_ for a certain time, the damage induced by Aβ_1–42_ may not be improved by withdrawing Aβ_1–42_. Furthermore, our results Figure 2A) suggest that the sirt-1 level decreased during Aβ_1–42_ incubation and then recovered when Aβ_1–42_ was withdrawn, which seems to be incompatible with the notion of damage memory; therefore, to address this issue, we assumed that the decreased Sirt-1 activity may be related to the NAD^+^ (Nicotinamide Adenine Dinucleotide) exhaustion, which is a pivotal cofactor of sirtuin-1 and influences Sirt-1 activity significantly. To test this hypothesis, the NAD^+^ level was determined by HPLC. The results of NAD^+^ show that the levels of NAD^+^ decreased continually in both the Aβ group and damage memory group, which suggests that the decreased Sirt-1 activity may be related to the NAD^+^ exhaustion ($p_{{NAD}^{+}\;}<0.05$, *n* = 6, $t_{{NAD}^{+}}\;=\;-4.97$). As the NAD^+^ level is related to mitochondria, it is essential to investigate how the mitochondria are involved in the formation of damage memory. The Sirt-1 related mitochondria factors p66^SHC^, ROS, and MMP were measured by Western blot and fluorescence commerce kits for ROS and MMP, respectively. The Western blot brands for p66^SHC^ are shown in Appendix A. The measurements show that MMP decreased, whereas p66^SHC^ and ROS increased continually in both the Aβ group and damage memory group. Compared with the control group, NAD^+^, p66^SHC^, ROS, Sirt-1 activity, and MMP had significant differences ($p_{p66^{SHC}}<0.05$, $t_{p66^{SHC}}\;=\;4.88$; $p_{MMP}<0.05$, $t_{MMP}\;=\;-7.24$; $p_{ROS}$ < 0.01, $t_{ROS\;}\;$= 4.57; $p_{sirt}<0.01$, $t_{sirt}\;=\;-6.33$, *n* = 6, *t*-test) in both the Aβ group and damage memory at 24 h; however, NAD^+^, p66^SHC^, ROS, Sirt-1 activity, and MMP had no significant difference ($p_{{NAD}^{+}\;}>0.05$, $t_{{NAD}^{+}}\;=\;1.50$; $p_{p66^{SHC}}>0.05$, $t_{p66^{SHC}}\;=\;0.08$; $p_{MMP}>0.05$, $t_{MMP}\;=\;0.98$; $p_{ROS}>0.05$, $t_{ROS}$ = −0.77; $p_{sirt}>0.05$, $t_{sirt}\;=\;-1.09$, *n* = 6, *t*-test) between the Aβ group and damage memory at 24 h. These results suggest that when the hCMEC/D3 cell is exposed to Aβ_1–42_ for 12 h, the levels of NAD^+^, p66^SHC^, ROS, and MMP may alter and are unable to recover to their baseline level. To determine whether intracellular Aβ accumulation is involved in the formation of damage memory, we performed Western blot analysis to measure the intracellular Aβ level after hCMEC/D3 cells were exposed to Aβ for 24 h. The results (the Western blot brands are given in Appendix A) showed that no intracellular Aβ accumulation was detected; therefore, it seems that intracellular Aβ accumulation may not be involved in the formation of damage memory.

### 2.2. Stimulating Sirt-1 Relieves the Endothelial Damage Memory

As the expression of sirt-1 was altered in the damage memory group compared to the control group, sirt-1 may affect the formation of the endothelial damage memory. There are two ways to stimulate the activity of sirt-1. First, a selective small molecule activator of sirt-1 SRT2104 is able to increase sirt-1 activity. Second, as NAD^+^ is a vital cofactor of sirt-1, NAD^+^ supplements can also stimulate the activity of sirt-1. Both of these methods were used to test whether the activity of sirt-1 may affect the formation of the endothelial damage memory. In the first experiment, a sirt-1 activator SRT2104 was used to stimulate sirt-1 activity. The results are shown in Figure 3A. Compared with the damage memory group, the levels of NAD^+^, MMP, Mn-SOD (Mn Superoxide Dismutase), and cell vitality in SRT2104 treated groups increased significantly ($p_{NAD^{+}\;}<0.05$, $t_{NAD^{+}\;}=\;17.35$; $p_{MMP}<0.05$, $t_{MMP}\;=\;11.29$; $p_{MTT}<0.05$, $t_{MTT}\;=\;31.53$; $p_{SOD}<0.01$, $t_{SOD}\;=\;10.41$, *n* = 6, *t*-test), whereas p66^SHC^ and ROS decreased significantly ($p_{p66}$ < 0.05, $t_{p66}$ = −11.01; $p_{ROS}<0.01$, $t_{ROS}\;=\;-5.04$, *n* = 6, *t*-test). The Western blot brands of p66^SHC^ and Mn-SOD are given in Appendix A, respectively. The variations of the levels of the above biomarkers are dose-dependent; therefore, our results suggest that stimulating sirt-1 relieves the endothelial damage memory.

In the second experiment, the NAD^+^ supplement was used to increase the activity of sirt-1. The results are shown in Figure 3C. Compared with the memory group, the levels of Mn-SOD, MMP, and cell vitality in NAD^+^ treated groups increased significantly ($p_{SOD}<0.05$, $t_{SOD}\;=\;2.68$; $p_{MMP}<0.05$, $t_{MMP}\;=\;3.61$; $p_{MTT}<0.05$, $t_{MTT}\;=\;9.11$, *n* = 6, *t*-test), whereas p66^SHC^ and ROS decreased significantly ($p_{p66}$ < 0.05, $t_{p66}$ = −8.94; $p_{ROS}<0.01$, $t_{ROS}\;=\;-11.89$
*n* = 6, *t*-test). The Western blot brands of p66^SHC^ and Mn-SOD are given in Appendix A, respectively. The variations of the levels of the above biomarkers are dose-dependent; therefore, our results suggest that the NAD^+^ supplement relieves the endothelial damage memory.

### 2.3. Inhibiting Sirt-1 Exacerbates the Endothelial Damage Memory

As stimulating sirt-1 may relieve the endothelial damage memory, presumably, inhibiting sirt-1 may have the opposite effect, and exacerbate the endothelial damage memory. To test this hypothesis, a sirt-1 inhibitor EX527 was used in a cell experiment. The results are shown in Figure 3B. Compared with the damage memory group, the levels of Mn-SOD, NAD^+^, MMP, and cell vitality in the EX527 treated groups decreased significantly ($p_{SOD}<0.01$, $t_{SOD}\;=\;-14.00$; $p_{NAD^{+}\;}<0.05$, $t_{NAD^{+}}\;=\;-29.89$, $p_{MMP}<0.05$, $t_{MMP}\;=\;-14.52$, $p_{MTT}<0.05$, $t_{MTT}\;=\;-40.11$, *n* = 6, *t*-test), whereas p66^SHC^ and ROS increased significantly ($p_{p66}$ < 0.05, $t_{p66}$ = 24.03; $p_{ROS}<0.01$, $t_{ROS}\;=\;39.27$, *n* = 6, *t*-test). The Western blot brands of p66^SHC^ and Mn-SOD are given in Appendix A, respectively. The variations of the levels of the above biomarkers are dose-dependent; therefore, our results suggest that inhibiting sirt-1 exacerbates the endothelial damage memory. The dysfunction of sirt-1 may not only increase the production of ROS but also impair the elimination of ROS. 

### 2.4. NAD^+^ and Sirt-1 Play Different Roles in the Dynamic Process of Endothelial Damage Memory

We questioned whether NAD^+^ and sirt-1 may play different roles in the endothelial damage memory kinetic process. To test this hypothesis, a mechanism-based kinetic progression model was developed. The visual predictive check (VPC) for this model is shown in Appendix A. The VPC plots show that the observed average data fall within the 95% prediction confidence interval. The bootstrapping values of estimated model parameters (Appendix A) remain near the estimation of the final parameters with a relatively low coefficient of variances (CV); therefore, the goodness of fit for the mechanism-based kinetic progression model is satisfactory.

After the internal validation of the mechanism-based kinetic progression was performed, simulations based on this model were conducted. The simulations were performed based on three scenarios. In this scenario, we aimed to investigate the time of endothelial damage memory formation. The relevance of this simulation was to provide baseline data for comparing the effects of different levels of sirt-1 and NAD^+^ on the time of endothelial damage memory formation. The results of the first simulation are shown in Appendix A. The results suggest that when the cells are treated with Aβ_1–42_ for more than 4 h, the levels of sirt-1, Sirt-1 activity, p66^SHC^, ROS, NAD^+^, MMP, and cell vitality may not be recovered by withdrawing Aβ_1–42_. In other words, the baseline of endothelial damage memory formation time might be 4 h post Aβ_1–42_ treatment in hCMEC/D3 cells. After the baseline time of the endothelial damage memory formation was determined, the simulation of the second scenario was performed. In this scenario, the level of sirt-1 or NAD^+^ changed and then the time of endothelial damage memory formation was estimated. The results of the simulation are shown in Figure 4. Changing the levels of both sirt-1 and NAD^+^ may alter the time of endothelial damage memory formation. Particularly, the variation of the endothelial damage memory formation time was found to be more sensitive to the changing of the sirt-1 level than that of the NAD^+^ level. In the third scenario, the methods for relieving the endothelial damage memory were investigated. The effects of the sirt-1 activator or NAD^+^ supplement on relieving the endothelial damage memory were estimated. The results of the above simulation are shown in Figure 5. When the cells are treated with the sirt-1 activator, the time of endothelial damage memory formation was delayed to 6 h post Aβ_1–42_ incubation. When the cells were treated with the NAD^+^ supplement, the time of endothelial damage memory formation was delayed to 8 h post Aβ_1–42_ incubation. These results suggest that the NAD^+^ supplement may be a potential method for delaying the formation of endothelial damage memory.

### 2.5. Different Roles of NAD^+^, and Sirt-1 in Delaying the Formation of Endothelial Damage Memory

The results of the simulations suggest that the NAD^+^ supplement may delay the formation of endothelial damage memory; cell experiments were performed to test this hypothesis. To test whether the NAD^+^ supplement can delay the formation of endothelial damage memory, the baseline time of endothelial damage memory formation should be determined. The results of the previous simulation suggest that the baseline time for endothelial damage memory formation may be 4 h after 2.5 µmol/mL Aβ_1–42_ incubation; therefore, the cell experiments were designed to investigate the baseline time of the damage memory formation and whether different Aβ_1–42_ concentrations affect the baseline time. The cell vitality in the 2 h memory group recovered, whereas it did not recover in the 4 h memory group after Aβ (2.5 µmol/mL) was withdrawn (Figure 6A). Compared with the Aβ (2.5 µmol/mL) group, the cell vitality in the 2 h memory group had significant difference (*p* < 0.05, *t* = 1.81, *t*-test) but it had no significant difference (*p* > 0.05, *t* = 1.48, *n* = 6, *t*-test) in 4 h memory group (Figure 6A). The results suggested that the baseline time of endothelial damage memory formation might be 4 h after Aβ_1–42_ (2.5 µmol/mL) incubation. In addition, to investigate the critical concentration and time exposure to Aβ, the formation time of the damage memory was estimated when the cells were incubated with different concentrations of Aβ. The results suggested that the formation time of the damage memory is Aβ concentration-dependent, as its curve fits the $E_{max}$ model (Figure 6B). We found that when the Aβ concentration increases, the damage memory forms earlier.

After the baseline time of endothelial damage memory formation was estimated, the effect of the sirt-1 activator or NAD^+^ supplement on delaying endothelial damage memory formation was investigated. The above simulation shows that the formation of endothelial damage memory may be delayed to 6 h or 8 h after Aβ_1–42_ incubation by SRT 2104 or NAD^+^ supplement, respectively. The cell experiments were designed according to the simulation. The results of the SRT2104 treatment are shown in Figure 7A. The results suggest that the cell vitality in the 4 h SRT2104 group (including low dose and high dose) is significantly higher (*p* < 0.05, *t* = 5.17, *n* = 6, *t*-test) than that in the 6 h SRT2014 (including low dose and high dose) group and damage memory group, whereas there is no significant difference (*p* > 0.05, *t* = −1.05, *n* = 6, *t*-test) between the 6 h SRT2104 (including low dose and high dose) group and damage memory group. Compared with the damage memory group, the levels of ROS and p66^SHC^ decreased significantly ($p_{ROS}<0.01$, $t_{ROS}\;=\;-11.89$; $p_{p66}$ < 0.05, $t_{p66}$ = −33.37, *n* = 6, *t*-test), whereas the levels of Mn-SOD, MMP, sirt-1 activity, and NAD^+^ increased significantly ($p_{SOD}<0.01$, $t_{SOD}\;=\;10.41$; $p_{MMP}<0.05$, $t_{MMP}\;=\;3.78$; $p_{{NAD}^{+}\;}<0.05$, $t_{{NAD}^{+}}\;=\;3.46$; $p_{sirt}<0.01$, $t_{sirt}\;=\;5.59$, *n* = 6, *t*-test) in SRT2104 treated cells; therefore, treating with SRT2104 may delay the endothelial damage memory formation to 6 h after Aβ_1–42_ incubation.

The results of NAD^+^ supplement treatment are shown in Figure 7B. The cell vitality in the 6 h NAD^+^ group (including low dose and high dose) is significantly higher (*p* < 0.05, *t* = 9.11, *n* = 6, *t*-test) than that in the 8 h NAD^+^ (including low dose and high dose) group and damage memory group, whereas there is no significant (*p* > 0.05, *t* = 2.11, *n* = 6, *t*-test) difference between the 8 h NAD^+^ (including low dose and high dose) group and damage memory group. Compared with the damage memory group, the levels of ROS and p66^SHC^ decreased significantly ($p_{ROS}<0.01$, $t_{ROS}\;=\;-11.89$; $p_{p66}$ < 0.05, $t_{p66}$ = −8.94, *n* = 6, *t*-test), whereas the levels of sirt-1 activity, Mn-SOD, and MMP increased significantly ($p_{sirt}<0.01$, $t_{sirt}\;=\;4.70$; $p_{SOD}<0.05$, $t_{SOD}\;=\;2.68$; $p_{p66}$ < 0.05, $t_{p66}$ = 3.61, *n* = 6, *t*-test) in NAD^+^ treated cells; therefore, treating with NAD^+^ may delay the endothelial damage memory formation to 8 h after Aβ_1–42_ incubation. The experiment results suggest that, compared to when combined with the sirt-1 activator, the NAD^+^ supplement may exhibit better effects on delaying the formation of endothelial damage memory (Figure 7C).

## 3. Discussion

Previous research has demonstrated that cerebrovascular endothelial cell damage is recognized as a contributor to the AD pathogenesis and Aβ may impair the cerebrovascular function [10,19]; however, AD animal model research has shown that the cerebrovascular function cannot be improved by removing Aβ [9,11,12]. In this study, we focused on whether the cerebral endothelium function can be repaired by removing Aβ in the early stage of AD progression. Acute Aβ exposure experiments were performed to simulate removing Aβ in the early stage of AD progression. Our results suggest that the brain vascular endothelial cells may remember the damage induced by Aβ exposure and their proliferative activity cannot be relieved after Aβ is withdrawn. In this study, a new feature, referred to as cerebrovascular endothelial cell damage memory, was introduced to explain these paradoxical results. 

Our results suggest that sirt-1 may be involved in the formation of cerebrovascular endothelial cell damage memory (Figure 8). sirt-1 is an NAD^+^ dependent protein deacetylase that occupies the cytoplasm and nucleus [20]. It may suppress gene transcription of the mitochondrial adaptor p66^SHC^ by deacetylating histone 3 binding to the p66^SHC^ promoter [21,22]. Inhibition of sirt-1 increased acetylated histone H3 binding to the p66^SHC^ promoter and induced the overexpression of p66^SHC^. The increased p66^SHC^ would open the mitochondrial permeability transition pore (PTP), which may result in the collapse of the mitochondrial membrane potential (MMP) [23]. When the PTP opens, the contact between the cytosolic and the mitochondrial pools of pyridine nucleotides may reduce NAD^+^ via enzymatic reactions [24]. According to the above research, a hypothesis for the mechanism of cerebrovascular endothelial damage memory was proposed: the sirt-1 level of the cerebrovascular endothelial cells may be decreased by Aβ exposure, then, the decreased sirt-1 could overexpress p66^SCH^, which may cause MMP collapse inducing NAD^+^ level reduction; NAD^+^ is a vital coenzyme of sirt-1, and low-level NAD^+^ may exacerbate sirt-1 deactivation, which then further reduces MMP, which may then form a vicious cycle. sirt-1 downregulation may be related to the ROS production induced by Aβ. Our results suggest that the damage memory may induce ROS accumulation, which is consistent with previous research. Previous clinical research has demonstrated a significant correlation between sirt-1 and Aβ levels in the brain (seen in human patients), and Aβ may suppress the expression of sirt-1 [17]; therefore, Aβ-induced ROS production may cause the depletion of sirt-1 expression [25,26].

Furthermore, the dynamic process of cerebrovascular endothelial cell damage memory formation was investigated using the mechanism-based kinetic progression model. According to our model, the progression of cerebrovascular endothelial cell damage memory might be divided into two phases. The first phase is the formation phase, which is defined as when the cell vitality can be recovered by removing Aβ_1–42_. The second phase, the maintenance phase, is defined as when the cell vitality cannot be recovered by removing Aβ_1–42_. The roles of sirt-1 and NAD^+^ are different in different phases. sirt-1 is an initiator in the formation phase. A decreased sirt-1 level may collapse the mitochondrial membrane potential, which may inhibit the production of NAD^+^. As NAD^+^ is exhausted, even if sirt-1 levels recover, it may not fully function as a histone deacetylase as it would lack the crucial cofactor NAD^+^. In other words, when the maintenance phase is reached, the lack of NAD^+^ may be an important factor to maintain the endothelial damage cycle. In summary, a decreased sirt-1 level is an initiator to activate the endothelial damage cycle. When the cycle is formed, it is maintained by low levels of NAD^+^ and the variation of sirt-1 level only has a limited impact on the damage cycle. Our experiments demonstrate that this cycle may induce mitochondria dysfunction and ROS accumulation. sirt-1, p66^SHC^, and Mn-SOD are affected by triggering this cycle. In this study, our results suggest that Aβ may suppress the expression of sirt-1, which may cause the overexpression of p66^SHC^. Increased p66^SHC^ may open the mitochondrial permeability transition pore (PTP), which may result in the collapse of the mitochondrial membrane potential (MMP). When the PTP opens, the contact between the cytosolic and the mitochondrial pools of pyridine nucleotides may reduce the NAD^+^ via enzymatic reactions. NAD^+^ is a vital coenzyme of SIRT-1, and low-level NAD^+^ may exacerbate the sirt-1 deactivation, then further reduce MMP, which may form a vicious cycle. Besides mitochondria dysfunction, this cycle may also induce ROS accumulation. The cycle may not only induce ROS production but also impair ROS elimination via the suppression of Mn-SOD expression. 

The above kinetic process analysis may provide insight into methods to improve the damage memory, which may contribute to reducing neuronal damage according to previous clinical research [27]. The results of our research suggest that both the sirt-1 activator and NAD^+^ supplement may exhibit endothelial protection effects. Previous research has demonstrated the roles of sirt-1 in AD pathology. The loss of sirt-1 is closely associated with the accumulation of amyloid-β and τ in the cerebral cortex of AD patients [17]. Research in cell culture and genetic mouse models has identified the potential protective role of sirt-1 activators against AD [28,29]. The sirt-1 activator SRT2104, similar to other sirt-1 activators, has been found to increase mitochondrial content and suppress the inflammation pathways [30]. It also exhibits endothelial protective effects, which were observed in this study [30]. Besides sirt-1 activators, NAD^+^ boosters or supplements may have potential endothelial protective effects. Clinical research has shown that stimulating the NAD^+^ metabolism in healthy middle-aged and older adults may reduce blood pressure and arterial stiffness [31]; however, the roles of the above treatments are still different in improving cerebrovascular endothelial cell damage memory. Compared with sirt-1 activator SRT2104, NAD^+^ supplement may have more potent effects on delaying the formation of cerebrovascular endothelial cell damage memory. This result may be due to the different roles of sirt-1 and NAD^+^ in the cerebrovascular endothelial cell damage memory dynamic process. sirt-1 may mainly play a role in initializing the cerebrovascular endothelial cell damage cycle. Once the cycle has formed, the variation of sirt-1 may have a limited impact on the cycle due to the lack of the vital cofactor NAD^+^; therefore, NAD^+^ supplements or boosters may be a potential method for improving the cerebrovascular endothelial cell damage memory. Furthermore, although our cell damage kinetics model was developed based on the cell model produced by Aβ_1–42_, previous research has suggested that the cytotoxicity of Aβ_1–40_ and Aβ_1–42_ are similar [32]; therefore, the structure of the proposed model can be applied for Aβ_1–40_ and the parameters of the proposed model may need to be further validated for Aβ_1–40_.

## 4. Method

### 4.1. Cell Culture

hCMEC/D3 cells were cultured in complete RPMI 1640 and seeded on glass coverslips in 12-well plates for ELISA and Western blotting, 24 well plates for HPLC and MMP assays, or 96-well plates for cell vitality and ROS assays. All cell lines were maintained at 37 °C and 5% CO_2_. Cell lines were validated by short tandem repeat (STR) profiling. The hCMEC/D3 were provided by Xiaodong Liu from the China Medicine University. 

### 4.2. Cell Treatment

Aβ peptide was used to prepare the AD cerebrovascular endothelial cell dysfunction model. The stock solution of Aβ peptide (100 μM) was prepared by dissolving 1 mg freeze-dried Aβ peptide powder in 2208 μL PBS (Phosphate Buffer Saline) and 45 μL DMSO. The stock solution was diluted to 2.5 μM with complete RPMI 1640 solution for in vitro model preparation. To investigate whether the cerebrovascular endothelial cell dysfunction memory exists in endothelial cells, the hCMEC/D3 cells were divided into three groups. For the first group (control group), the hCMEC/D3 cells were cultured in complete RPMI 1640. For the second group (the Aβ group), the hCMEC/D3 cells were incubated with complete RPMI 1640 containing 2.5 μM Aβ peptide for 24 h. For the third group (the damage memory group), the hCMEC/D3 cells were incubated with 2.5 μM Aβ peptide for 12 h and then the Aβ was withdrawn for another 12 h incubation. For all three groups, the cell samples were collected at 0, 2, 4, 6, 8, 10, 12, 14, 16, 18, 20, 22, and 24 h for sirt-1, p66^SHC^, NAD^+^, reactive oxygen species (ROS), MMP, and cell vitality measurement. For mechanistic experiments, the cells in the damage memory group were treated with EX527 (a selective sirt-1 inhibitor), SRT2104 (a selective sirt-1 activator), and NAD^+^ [33,34].

### 4.3. ELISA Kit

The expression of sirt-1 was measured by a commercial ELISA kit obtained from Abcam (Cambridge, UK).

### 4.4. Immunoblot

The cells were lysed in an RIPA buffer and quantified using a BCA assay. Equal amounts of total protein were separated by SDS-PAGE followed by electrophoretic transfer to polyvinylidene fluoride (PVDF) membranes (Millipore, Burlington, MA, USA). After blocking membranes for 1 h with 5% skim milk powder in PBST, p66^SHC^ anti-body, amyloid-β anti-body, or Mn-SOD anti-body were immunodetected by incubating for 16 h in primary antibody in blocking buffers. Membranes were washed extensively with PBST or TBST and incubated with anti-rabbit secondary anti-body in blocking buffer. After 1 h, membranes were washed as above and developed using enhanced chemiluminescence. Densitometric images were captured with ImageJ and band intensity normalized to the control group.

### 4.5. Cell Vitality Assay

To evaluate the vitality of cells, the growth medium was disposed of. Then, we washed the cells twice with PBS. A 150 μL 0.5 mg/mL MTT solution was added to each well of the 96-well plates. After incubation for 90 min at 37 °C with MTT, the supernatant in each well was removed. The precipitated formazan was solubilized with DMSO and quantified spectrophotometrically at 550 nm.

### 4.6. MMP Assay

Following incubation with the JC-1 at 37 °C/45 min, the culture medium was removed, and plates were washed with PBS. Finally, fluorescence was measured in a Perkin Elmer LS-50B fluorescence microplate reader set at 525 nm (excitation) and 590 nm (emission).

### 4.7. ROS Measurement

Following incubation with the DCFH-DA (10 µmol/L) at 37 °C/20 min, the culture medium was removed and plates were washed with PBS. Finally, fluorescence was measured in a Perkin Elmer LS-50B fluorescence microplate reader set at 488 nm (excitation) and 525 nm (emission).

### 4.8. NAD^+^ Sample Preparation and HPLC Condition

The cells were lysed by freeze–thaw cycles. The cell extract was centrifuged for 10 min at 15,000 rpm under 4 °C. A total of 100 μL of the supernatant was stored at −70 °C until analysis.

The prepared samples are analyzed by an HPLC method according to the previous research with slight modifications [35]. The prepared samples were injected into an Agilent ZORBAX SB-Aq column (5.0 μm, 150 × 2.0 mm). The mobile phase consisted of water containing 25 g/L Na_2_HPO_4_·12H_2_O. The flow rate of the mobile phase was 1 mL∙min^−1^. The injection volume was 20 μL. The column oven was conditioned at +40 °C and UV detection was set to 210 nm. 

### 4.9. Sirt-1 Activity Assay

The sirt-1 activity was measured by a commercial kit as instructed by the manufacturer (obtained from Yanyu Biotech, Shanghai, China). The Cell Lysis solution was incubated with 5 μL of fluorescence substrate (100 µmol/L) and NAD^+^ (100 µmol/L) for 30 min at 37 °C. The fluorescence was subsequently monitored for 30 min at 360 nm (excitation) and 460 nm (emission).

### 4.10. Mechanism-Based Kinetic Progression Model Development

In this study, the data for model development were collected in the above cell experiment. After the model was developed, simulations were performed to investigate the dynamic characters of cerebrovascular endothelial damage memory. The mechanism-based kinetic progression model is composed of five linked turn over equations:

The basal equation of sirt-1 level (Equation (1)) is depicted by a zero-order production rate ($k_{in}^{SIRT1}$) and a first-order degradation rate ($k_{out}^{SIRT1}$). In this system, the expression of sirt-1 may be inhibited by Aβ whose concentration is constant in the cell experiments; therefore, inhibition of sirt-1 expression induced by Aβ is assumed to be constant, which is described by parameter $E_{A\beta }$. When the cell is incubated in growth medium without Aβ, the value of $E_{A\beta }$ is set to 1. The basal equation of p66^SHC^ level (Equation (2)) is depicted by a zero-order production rate ($k_{in}^{p66^{SHC}}$) and a first-order degradation rate ($k_{out}^{p66^{SHC}}$). sirt-1 may decrease p66^SHC^ expression. Additionally, NAD^+^ is the cofactor of sirt-1, which may also affect the expression of p66^SHC^. The effects of sirt-1 and NAD^+^ on p66^SHC^ expression are assumed to be linear which are described by parameters $k_{p66^{SHC}}^{SIRT1}$ and $k_{p66^{SHC}}^{NAD^{+}}$, respectively. The basal equation of ROS level (Equation (2)) is depicted by a zero-order production rate ($k_{in}^{ROS}$) and a first-order degradation rate ($k_{out}^{ROS}$). The effect of p66^SHC^ on rising ROS level is described by $E_{max}$ model containing two parameters $E_{max}^{p66^{SHC}}$ and $EC_{50}^{p66^{SHC}}$. The basal equation of NAD^+^ level (Equation (3)) is depicted by a zero-order production rate ($k_{in}^{NAD^{+}}$) and a first-order degradation rate ($k_{out}^{NAD^{+}}$). The basal equation of MMP level (Equation (4)) is depicted by a zero-order production rate ($k_{in}^{MMP}$) and a first-order degradation rate ($k_{out}^{MMP}$). The MMP collapse induced by ROS is assumed to be described by a $E_{max}$ model that contains two parameters: $E_{max}^{ROS}$ and $EC_{50}^{ROS}$. The basal equation of cell vitality (Equation (5)) is depicted by a zero-order production rate ($k_{in}^{MTT}$) and a first-order degradation rate ($k_{out}^{MTT}$). The cell vitality may be affected by MMP, whose effect is assumed to be described by an $E_{max}$ model including two parameters $E_{max}^{MMP}$ and $EC_{50}^{MMP}$.
(1)$$\frac{dc_{SIRT1}}{dt}=k_{in}^{SIRT1}E_{A\beta }-k_{out}^{SIRT1}c_{SIRT1}$$
(2)$$\frac{dc_{p66^{SHC}}}{dt}=k_{in}^{p66^{SHC}}\left(1-k_{p66^{SHC}}^{SIRT1}c_{SIRT1}-k_{p66^{SHC}}^{NAD^{+}}c_{NAD^{+}}\right)-k_{out}^{p66^{SHC}}c_{p66^{SHC}}$$
(3)$$\frac{dc_{ROS}}{dt}=k_{in}^{ROS}\cdot \frac{E_{max}^{p66^{SHC}}c_{p66^{SHC}}}{EC_{50}^{p66^{SHC}}+c_{p66^{SHC}}}-k_{out}^{ROS}c_{ROS}$$
(4)$$\frac{dc_{NAD^{+}}}{dt}=k_{in}^{NAD^{+}}c_{MMP}-k_{out}^{NAD^{+}}c_{NAD^{+}}$$
(5)$$\frac{dc_{MMP}}{dt}=k_{in}^{MMP}-k_{out}^{MMP}\left(c_{MMP}+\frac{E_{max}^{ROS}c_{ROS}}{EC_{50}^{ROS}+c_{ROS}}\right)$$
(6)$$\frac{dc_{MTT}}{dt}=k_{in}^{MTT}\left(1+\frac{E_{max}^{MMP}c_{MMP}}{EC_{50}^{MMP}+c_{MMP}}\right)-k_{out}^{MTT}c_{MTT}$$

### 4.11. Simulation

The simulation can provide insight into three issues. First, the simulation can help find the time of cerebrovascular endothelial damage memory formation. For this scenario, Aβ was withdrawn at different time points and levels of sirt-1, p66^SHC^, NAD^+^, MMP, and cell vitality were estimated to find the time at which the cell vitality may recover after was withdrawn. The formation of cerebrovascular endothelial damage memory was defined as when the cell vitality decreases more than 30% compared with the control group and when it cannot recover after Aβ is withdrawn. Second, the influence factors for cerebrovascular endothelial damage memory formation were investigated by simulations. In this scenario, the effects of levels of sirt-1 and its cofactor NAD^+^ on cerebrovascular endothelial damage memory formation were investigated. When the sirt-1 level or NAD^+^ level changes, the time for cerebrovascular endothelial damage memory formation was estimated. Third, the methods for delaying the formation of endothelial damage memory were investigated by simulation. In this scenario, the endothelial improvement effect of sirt-1 activator and NAD^+^ supplement were simulated. The improvement method proposed by simulation was validated in cell experiments. 

### 4.12. Simulation Validation

To validate the simulation based on the kinetic progression, three cell experiments were performed. The purpose of the first cell experiment was to validate the baseline time of the formation of endothelial damage memory (Figure 9A). Validation of the baseline time could help us to compare the different endothelial function improvement methods. In the first experiment, the hCMEC/D3 cells were divided into four groups. In the first group (control group), the cells were incubated with complete RPMI 1640 for 24 h. In the second group (2 h memory group), the cells were incubated with a culture medium containing Aβ_1–42_ for 2 h, and then we withdrew Aβ_1–42_ for another 22 h incubation period. In the third group (4 h memory group), the cells were incubated with a culture medium containing Aβ_1–42_ for 4 h and then we withdrew Aβ_1–42_ for another 20 h incubation. In the fourth group (Aβ group), the cells were incubated with a culture medium containing Aβ_1–42_ for 24 h. All cell samples were collected for cell vitality measurement after 24 h incubation. 

The purpose of the second experiment was to investigate the effects of sirt-1 activator SRT2104 on delaying the formation of endothelial damage memory (Figure 9B). In this experiment, the cells were divided into seven groups. In the first group (control group), the cells were incubated with complete RPMI 1640 for 24 h. In the second group (Aβ group), the cells were incubated with Aβ for 24 h. In the third group (damage memory group), the protocol was the same as the 4 h memory group in the first experiment. In the fourth and fifth groups (4 h high and low dose SRT2104 groups), the cells were incubated with a culture medium containing Aβ_1–42_ for 4 h and then we withdrew Aβ_1–42_ for another 20 h incubation, meanwhile, the cells were treated with 2 μmol/L and 1 μmol/L SRT2104, respectively, during the entire incubation. In the sixth and seventh groups (6 h high and low dose SRT2104 group), the cells were incubated with a culture medium containing Aβ_1–42_ for 6 h and then we withdrew ${A\beta }_{1-42}$ for another 18 h incubation, meanwhile, the cells were treated with 2 μmol/L and 1 μmol/L SRT2104, respectively, during the entire incubation. 

The purpose of the third experiment was to investigate the effects of the sirt-1 activator on delaying the formation of endothelial damage memory (Figure 9C). In this experiment, the cells were divided into seven groups. The protocols of the first group (control group) and second (Aβ group) are the same as those groups in the second experiment. In the third group (damage memory group), the cells were incubated with a culture medium containing Aβ_1–42_ for 6 h and then we withdrew Aβ for another 18 h incubation. In the fourth and fifth groups (6 h high and low dose NAD^+^ groups), the cells were incubated with a culture medium containing Aβ_1–42_ for 6 h and then we withdrew Aβ_1–42_ for another 18 h incubation; the cells were treated with 5 mmol/L and 1 mmol/L NAD^+^, respectively, during the entire incubation. In the sixth and seventh groups (8 h high and low dose NAD^+^ groups), the cells were incubated with a culture medium containing Aβ_1–42_ for 8 h and then we withdrew Aβ_1–42_ for another 16 h incubation; the cells were treated with 5 mmol/L and 1 mmol/L NAD^+^, respectively, during the entire incubation. All the cell samples were collected after 24 h incubation for p66^SHC^, NAD^+^, MMP, and cell vitality tests.

## 5. Conclusions

In this study, the kinetic progression of the vicious circle of cerebrovascular endothelial cell damage memory was demonstrated. sirt-1 is an initiator that activates the above-described cycle. Once the cycle is formed, it is maintained by a low level of NAD^+^, which suggests that NAD^+^ supplements may be a potential method for improving the cerebrovascular endothelial cell damage memory. The present study provides new insight into cerebrovascular endothelial damage in AD progression.

## Figures and Tables

**Figure 1 ijms-21-08226-f001:**
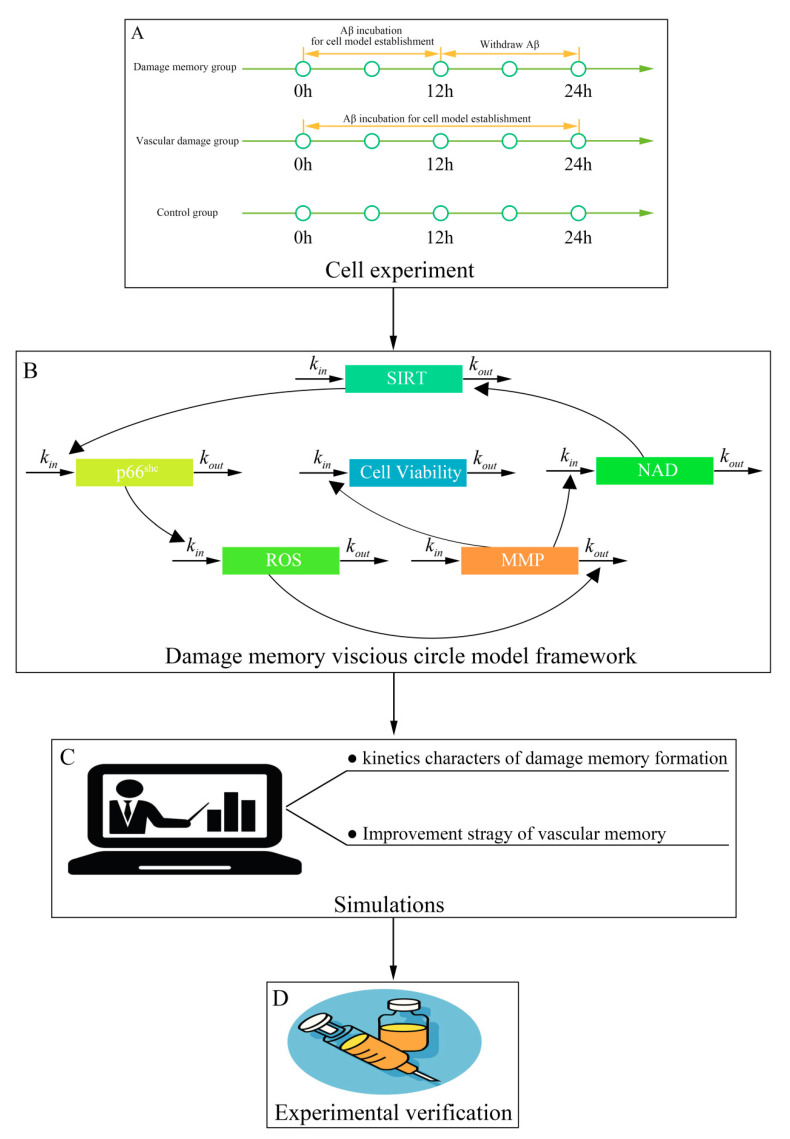
The framework of this study. (**A**) The procedure of in vitro experiments. (**B**) The schematic diagram for the kinetic model of damage memory. (**C**) The procedure for simulations. (**D**) Experimental validation for simulation.

**Figure 2 ijms-21-08226-f002:**
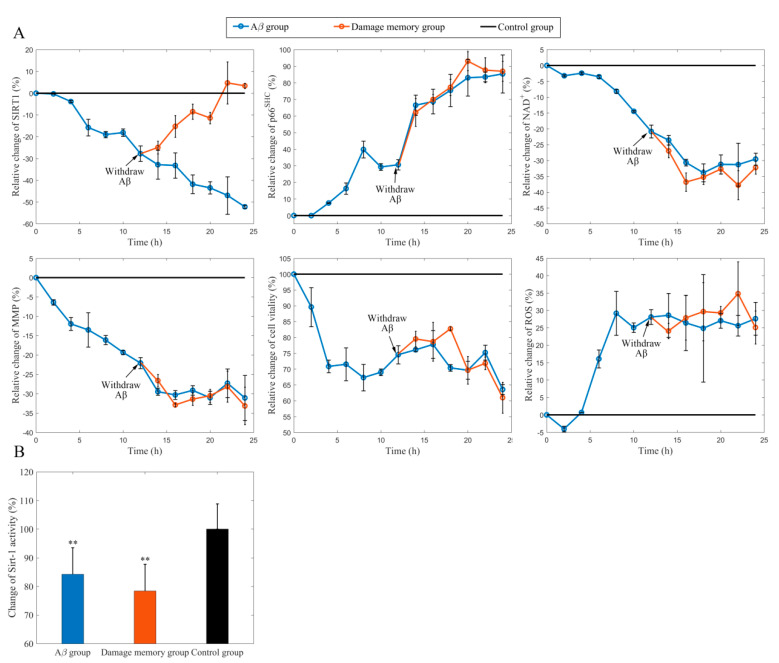
(**A**): The time course of the relative change of sirt-1, p66^SHC^, NAD^+^ (Nicotinamide Adenine Dinucleotide), reactive oxygen species (ROS), and mitochondrial membrane potential (MMP), and cell vitality compared to the control group. The black line represents the control group level, which is normalized to 100%. The blue line represents the β-amyloid (Aβ) group level. The red line represents the damage memory group. (**B**): The levels of sirt-1 activity in damage memory group, Aβ group, and control group, respectively. The error bar was generated with the mean ±SD. The difference between groups are tested by *t*-test (*n* = 6). ** *p* < 0.01.

**Figure 3 ijms-21-08226-f003:**
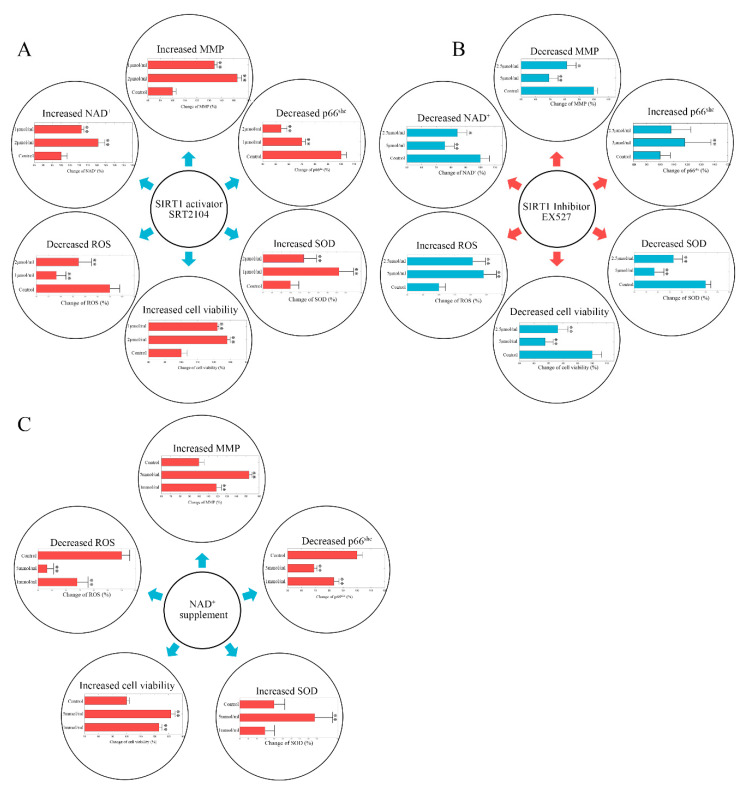
(**A**) The change of p66^SHC^, NAD^+^, MMP, ROS, Mn-SOD (Mn Superoxide Dismutase), and cell vitality in SRT2104 treated hCMEC/D3 cells. The control group data are normalized to 100%. (**B**) The change of p66^SHC^, NAD^+^, MMP, ROS, Mn-SOD, and cell vitality in EX527 treated hCMEC/D3 cells. The control group data are normalized to 100%. (**C**) The change of p66^SHC^, MMP, ROS, Mn-SOD, and cell vitality in NAD^+^ supplement treated hCMEC/D3 cells. The control group data are normalized to 100%. The difference between different groups was compared by *t*-test (*n* = 6). ** *p* < 0.01 * *p* < 0.05.

**Figure 4 ijms-21-08226-f004:**
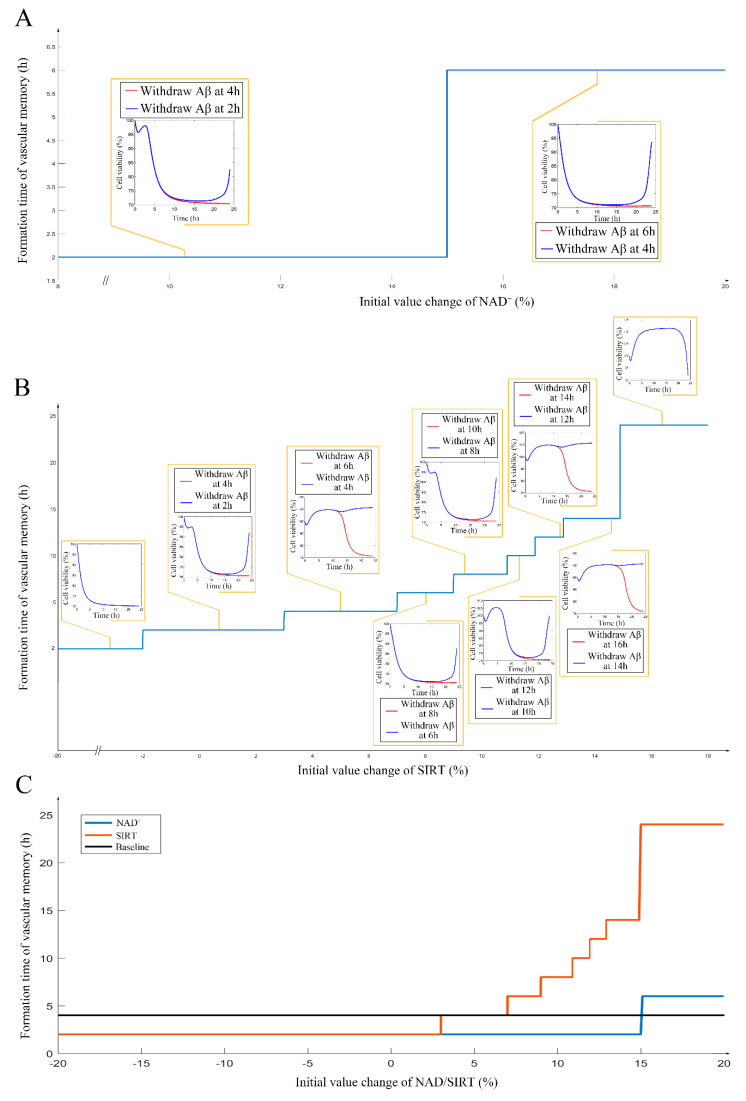
(**A**): The impact of different levels of NAD^+^ on the cerebrovascular endothelial cell damage memory formation time. (**B**): The impact of different levels of sirt-1 on the cerebrovascular endothelial cell damage memory formation time. (**C**): the summary plot of the impact of different levels of sirt-1 and NAD^+^ on the cerebrovascular endothelial cell damage memory formation time.

**Figure 5 ijms-21-08226-f005:**
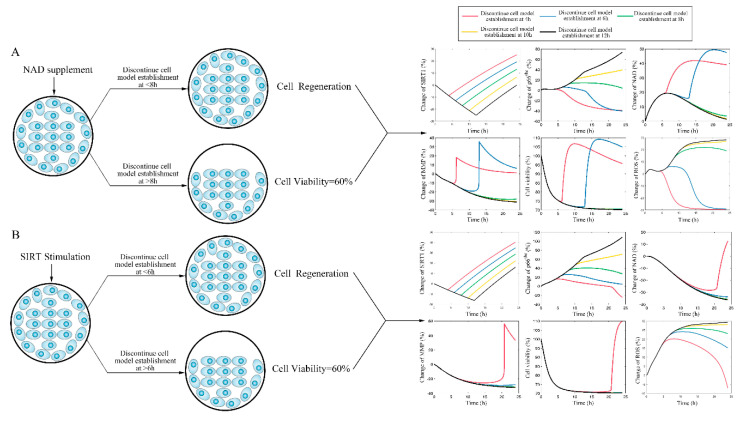
(**A**): The simulation for change of sirt-1, p66^SHC^, ROS, NAD^+^, MMP, and cell vitality in NAD^+^ supplement treated cells. (**B**): The simulation for change of sirt-1, p66^SHC^, ROS, NAD^+^, MMP, and cell vitality in the sirt-1 activator treated cells.

**Figure 6 ijms-21-08226-f006:**
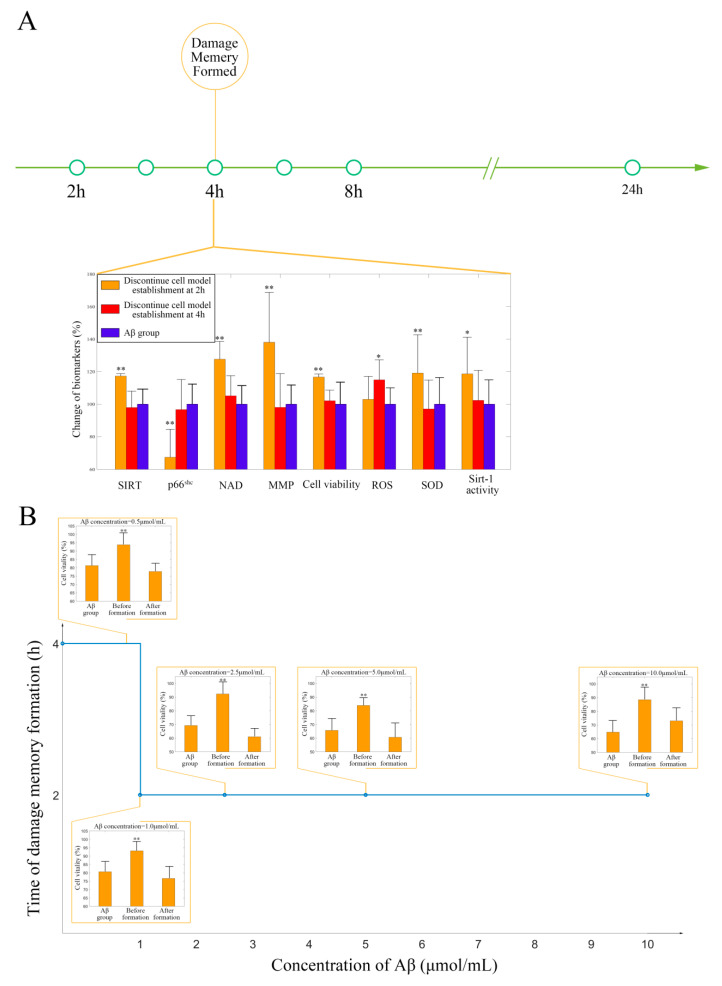
(**A**) The experimental validation of cerebrovascular endothelial cells’ damage memory formation time. (**B**) The damage memory formation time with different concentrations of Aβ incubation. The difference between the groups is compared by *t*-test (*n* = 6, the detailed results are shown in the main text). ** *p* < 0.01 * *p* < 0.05.

**Figure 7 ijms-21-08226-f007:**
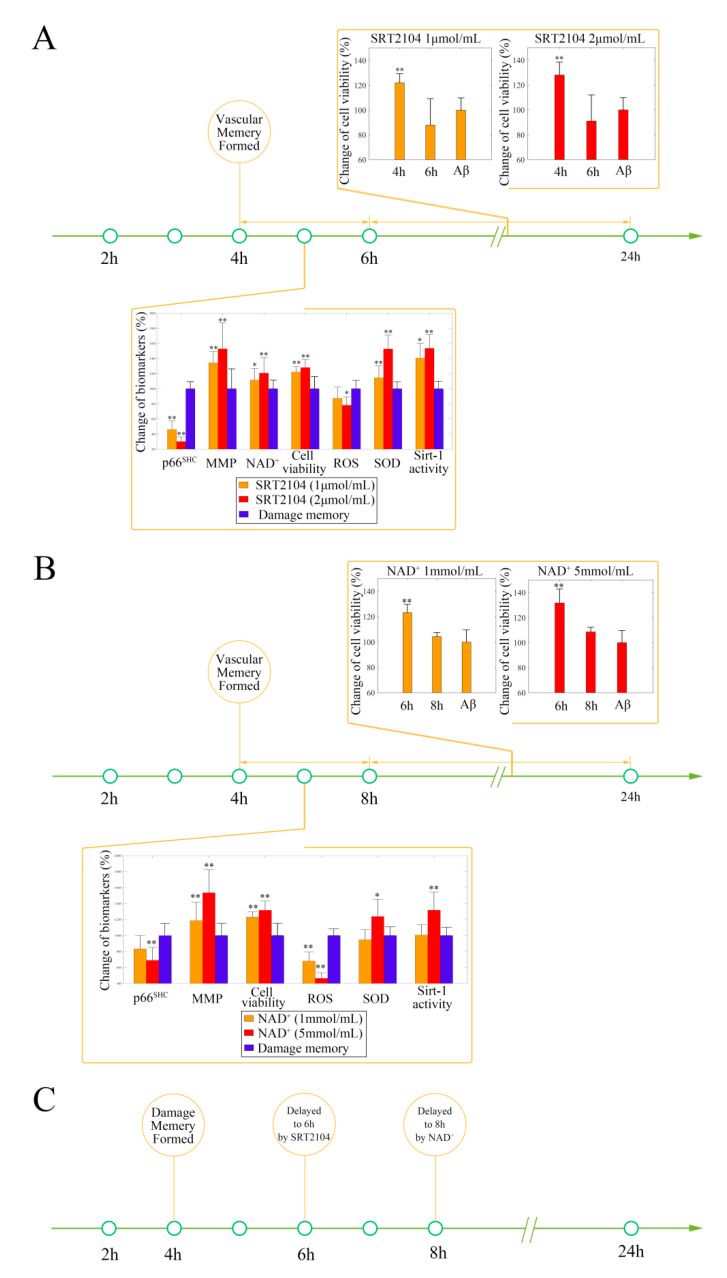
(**A**) The effect of SRT2104 on delaying the formation of cerebrovascular endothelial cells’ damage memory formation. (**B**) The effect of NAD^+^ supplement on delaying the formation of cerebrovascular endothelial cells’ damage memory formation. (**C**) Summary of the effect of SRT2104 and the NAD^+^ supplement on delaying the formation of cerebrovascular endothelial cell damage memory. The difference between groups is compared by *t*-test (*n* = 6, the detailed results are shown in the main text). ** *p* < 0.01 * *p* < 0.05.

**Figure 8 ijms-21-08226-f008:**
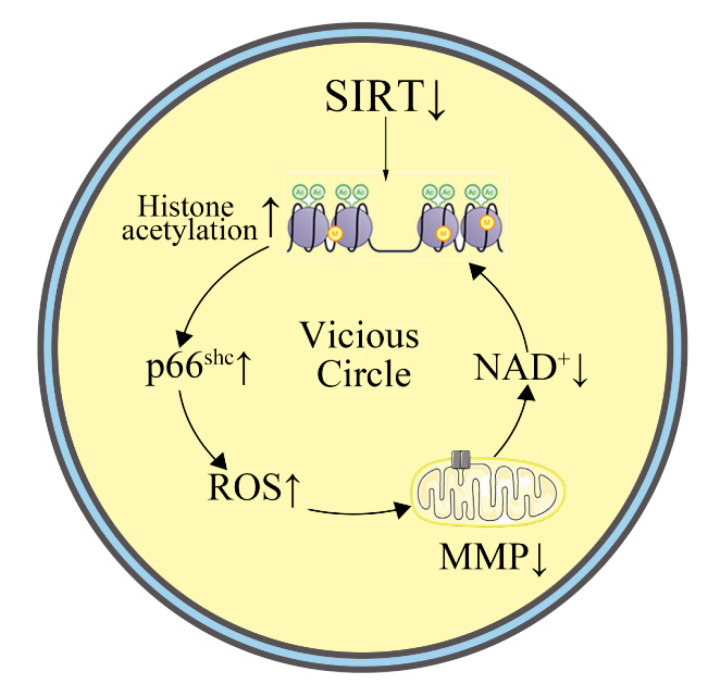
The cerebrovascular endothelial cell damage memory vicious circle.

**Figure 9 ijms-21-08226-f009:**
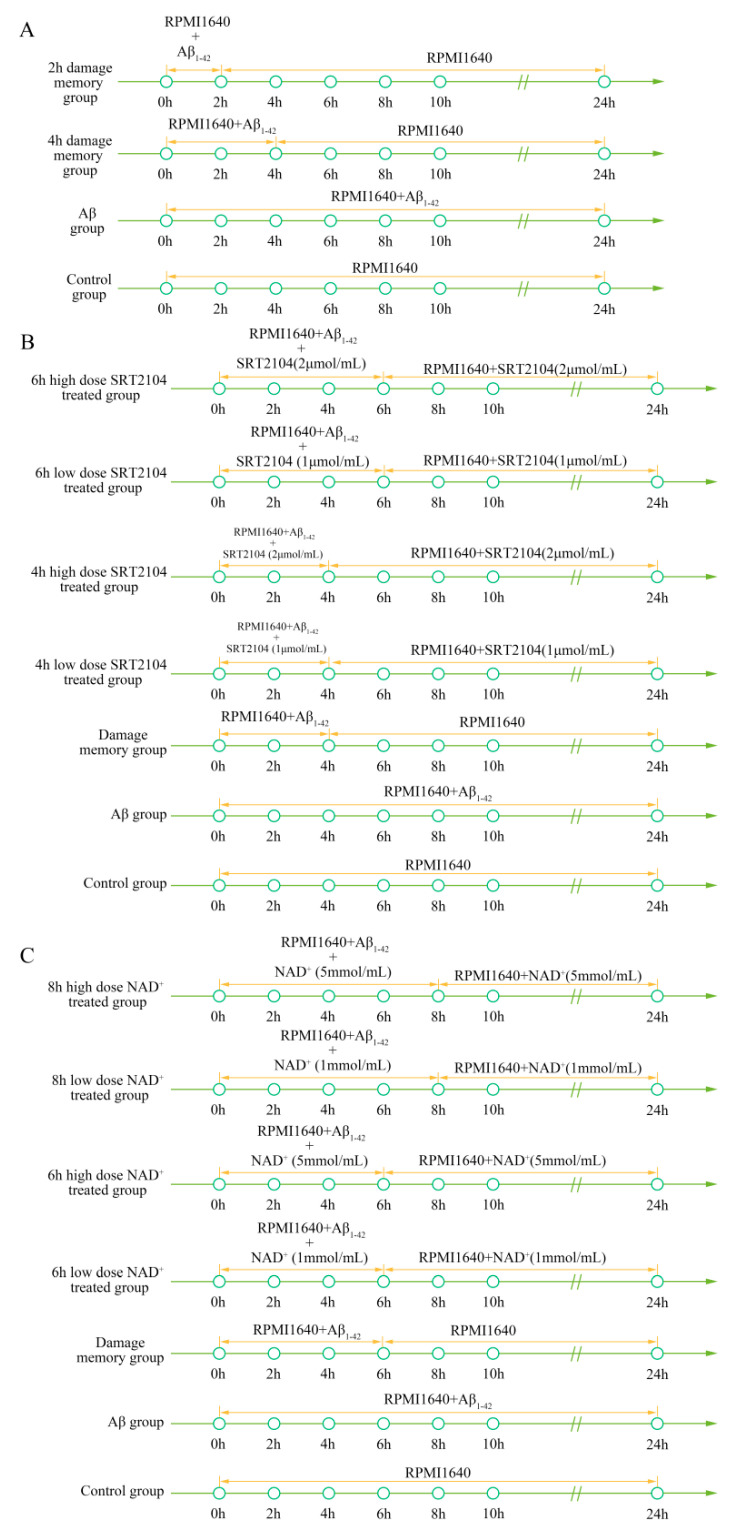
(**A**) Experimental protocol for determining the time of cerebrovascular endothelial cell damage memory formation. (**B**) Experimental protocol to evaluate the effect of SRT2104 on delaying the formation of cerebrovascular endothelial cell damage memory. (**C**) Experimental protocol to evaluate the effect of the NAD^+^ supplement on delaying the formation of cerebrovascular endothelial cell damage memory.

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
