# Peer review of "Aβ-Induced Damage Memory in hCMEC/D3 Cells Mediated by Sirtuin-1"

_ijms, 2020, doi:10.3390/ijms21218226_

Round 1
Reviewer 1 Report
The manuscript “Aβ-induced damage memory in hCMEC/D3 cell mediated by sirtuin-1” addresses a very interesting issue in AD, i.e. whether cerebrovascular endothelial memory of a damage induced by Aβ exists in endothelial cells. This is very important to understand the role of endothelial dysfunction during AD progression that can explain the failure of anti-bodies targeted at Aβ in clinical trials and it can be linked to the metabolic memory phenomenon. Furthermore, considering the importance of epigenetic dysfunction in AD, the role of the HDAC sirtuin-1 has been investigated.
In the Results section, par. 2.1 (pag. 2, line 81), it should be better to clarify which endothelial cell line is used in the experiment and why this cellular model has been chosen. Furthermore, it should be more clear for the readers whether authors clearly explain the reasons why NAD+, p66SHC, ROS and MMP are measured. While the role of sirtuin-1 has been clarified in the introduction, nothing has been explained before regarding these factors. In addition, in the text it should be shortly explained how cell vitality and the measure of the other factors has been evaluated. In the figure 1 legends, more details on the statistical analysis performed and on the way in which statistically significant differences have been represented should be added, including the n of each experiments.
Pag. 3, lane 114: the sentence “The results are shown in Fig4”, I assume it should be ““The results are shown in Fig2”. Similarly, at pag. 4, lane 138, the sentence “The results are shown in Fig 4B”, probably is “The results are shown in Fig 2B”. In both cases, authors should clarify. The way in which authors represented the data in figure 2 is absolutely not clear and it should be changed, making the results more clear for the readers. Even in the case of the figure 2 legend, the statistics is not clear, as for figure 2.
Table 1, figures 3 and 4 with the explanation of the data should be moved to the supplementary results, because in the main text it makes too complex the reading of the manuscript.
Pag. 5, lane 161: the sentence “The results of the first simulation are shown in the Figure 4”, but probably it’s figure 3.
Pag 6, lanes 173-174, there is sentence in place of a figure number, as well as in several lines of pages 9 and 10.
Figure 5, 6, 7 and 8 completely lack statistical analysis that should be described in the text, in the legends and included in the images, together with the n of the experiment.
The experimental models of figure 7 and 8 should be moved to the supplementary materials, so the graphs with the obtained results would be easier to read.
The images of western blots in supplementary materials have never been described in the main text, so their role and significance is unclear.
Author Response
1.In the Results section, par. 2.1 (pag. 2, line 81), it should be better to clarify which endothelial cell line is used in the experiment and why this cellular model has been chosen. Furthermore, it should be more clear for the readers whether authors clearly explain the reasons why NAD+, p66SHC, ROS and MMP are measured. While the role of sirtuin-1 has been clarified in the introduction, nothing has been explained before regarding these factors. In addition, in the text it should be shortly explained how cell vitality and the measure of the other factors has been evaluated. In the figure 1 legends, more details on the statistical analysis performed and on the way in which statistically significant differences have been represented should be added, including the n of each experiments.
Response: the aim of this study is to investigate whether the damage memory exists in brain endothelial cell. The hCMEC cell is a well established in vitro cerebral endothelial model which has been used in many previous researches. Therefore, hCMEC is selected in this study. We have added this statement in the results section of the revised manuscript. Measurement of NAD+ contributed to illustrating the formation of the damage memory. Our results showed that the level of Sirt-1 recovered after Aβ was withdrawn. But the activity of Sirt-1 still declined after Aβ was withdrawn. To explain these contradiction results, we assumed that the decreased Sirt-1 activity may be related to the NAD+ exhaustion which is a pivotal cofactor of sirtuin-1 and influences Sirt-1 activity significantly. To test this hypothesis, we determined the NAD+ level which suggested that the NAD+ exhaustion may be related to the decreased Sirt-1 activity.
The measurement of p66SHC, ROS and MMP contributed to illustrating the role of mitochondria dysfunction in damage memory formation. Decreased Sirt-1 activity may increase acetylated histone H3 binding to the p66SHC promoter and induce overexpression of p66SHC. The increased p66SHC would increase the ROS level and open the mitochondrial permeability transition pore (PTP) which may result into the collapse of the mitochondrial membrane potential (MMP). When the PTP opens, the contact between the cytosolic and the mitochondrial pools of pyridine nucleotides
may reduce NAD+ via enzymatic reactions which may further impair the activity of Sirt-1 and form the damage memory vicious circle. The above discussion have been added to the introduction section of revised manuscript.
The cell vitality was measured by MTT assay. NAD+ was determined by HPLC. p66SHC was determined by western blot. Sirt-1 was measured by ELISA kit. ROS and MMP were measured by commercial fluorescence kit respectively. The above analysis methods have been added to the revised manuscript. The error bar in the figure 1 was generated with mean±SD. The difference between groups are tested by t-test (n=6). The above details on the statistical analysis
have been added to the revised manuscript.
2. Pag. 3, lane 114: the sentence “The results are shown in Fig4”, I assume it should be ““The results are shown in Fig2”. Similarly, at pag. 4, lane 138, the sentence “The results are shown in Fig 4B”, probably is “The results are shown in Fig 2B”. In both cases, authors should clarify. The way in which authors represented the data in figure 2 is absolutely not clear and it should be changed, making the results more clear for the readers. Even in the case of the figure 2 legend, the statistics is not clear, as for figure 2.
Response: thank you for your advice. We have corrected the typos as your comments in the revised manuscript. The figure 2 is redrawn in the revised manuscript. The statistics has been added to the figure 2 legend in the revised manuscript.
3. Table 1, figures 3 and 4 with the explanation of the data should be moved to the supplementary results, because in the main text it makes too complex the reading of the manuscript.
Response: thank you for your advice. We have moved the table1 and figure 3, 4 to the
supplementary results in the revised manuscript.
4. Pag. 5, lane 161: the sentence “The results of the first simulation are shown in the Figure 4”, but probably it’s figure 3.
Response: thank you for your advice. The results of the first simulation is shown in the figure 4.
5. Pag 6, lanes 173-174, there is sentence in place of a figure number, as well as in several lines of pages 9 and 10.
Response: thank you for your advice. We have corrected these typos in the revised manuscript.
6. Figure 5, 6, 7 and 8 completely lack statistical analysis that should be described in the text, in the legends and included in the images, together with the n of the experiment.
Response: thank you for your advice. The figures 5 and 6 represent the results of simulation based on the kinetics model. The statistical procedures have been described in the method section. We have added the statistical analysis procedures to the legends of figure 7 and 8.
7. The experimental models of figure 7 and 8 should be moved to the supplementary materials, so the graphs with the obtained results would be easier to read.
Response: thank you for your advice. We have revised figure 7 and 8 as your comments.
8. The images of western blots in supplementary materials have never been described in
the main text, so their role and significance is unclear.
Response: thank you for your advice. These images of western blots are the original data for p66shc and Mn-SOD which have been described in the main text. These original images are the provided as the policy of the journal. We have described these images in the main text.
Reviewer 2 Report
In the present manuscript, the authors present a series of experiments (both in culture and using computational modeling) to examine the role of AB in endothelial cell damage associated with pathology in Alzheimer’s disease. Finding that removal of AB plaques in AD does not improve damage to endothelial cells, they aim to find the mechanism through which this long-lasting impairment acts, ultimately landing on changes in sirt-1 activity triggering and maintaining this damage. In line with this, they demonstrated the long-lasting damage to endothelial cells could be reduced by NAD treatments or sirt-1 activators.
Overall, the logic of the experiments is sound and I enjoyed the detailed mechanistic work. However, several aspects of the paper make it difficult to read. These can be easily addressed in a revision.
Major Comments:
The authors should address how the time frame in the current experiments might not necessarily be reflective of the type of impairments seen in human pathologies.
The logic of the designs should be laid out before the methods in this case, as the methods are at the end of the paper. Instead, I’d suggest going through a brief overview of what will be done in the paper. For example, lay out the hypotheses then say, “To test this, we…” (Moving Figure 10 up in the paper would help in this regard considerably.)
Statistics should be reported fully (e.g., even t-tests need associated degrees of freedom).
The paper is written in a way that suggests very little proof-reading was conducted. Above standard typos, which are usually not a huge issue to me, there are misplaced words, extraneous or missing words in some sentences, and several places just say “Error! Reference source not found!” in bolded font. Rather than walk through each of these individually, I would suggest a thorough examination of the manuscript before resubmitting (including figure captions, where many of these issues arise as well).
Minor comments:
Do the authors mean ‘vicious cycle’ rather than ‘vicious circle’? I would also suggest removing this term from the figures and instead just saying this triggers a cycle, rather than a vicious cycle.
I would give a more thorough, albeit still brief, introduction to the role of epigenetics in AD symptomology, rather than just saying that changes are observed.
Author Response
1. The authors should address how the time frame in the current experiments might not necessarily be reflective of the type of impairments seen in human pathologies.
Response: thank you for your advise. In this study, we focus on the issue that whether the cerebral endothelium function can be repaired by removing Aβ in early stage of AD progression. Therefore, acute Aβ exposure experiments were performed to simulate removing Aβ in early stage of AD progression. Our results suggested that the brain vascular endothelial cells may remember the damage induced by Aβ exposure and their proliferative activity can not relief after Aβ is withdrawn. We have make it clear in the revised manuscript.
2. The logic of the designs should be laid out before the methods in this case, as the methods are at the end of the paper. Instead, I’d suggest going through a brief overview of what will be done in the paper. For example, lay out the hypotheses then say, “To test this, we…” (Moving Figure 10 up in the paper would help in this regard considerably.)
Response: thank you for your advice. This study contains four steps. Firstly, cell experiments are performed to investigate that whether the damage memory exists in endothelial cells and obtain the data for the kinetics process of cerebrovascular endothelial cells damage. Secondly, a mathematical model is developed to describe the above kinetics process. Thirdly, simulations based on the above model are performed to investigate the kinetic characters of the damage process and improvement method of cerebrovascular endothelial cells damage. Fourthly, the improvement method proposed by the above simulations are validated by cell experiments. We have added the above brief overview to the revised manuscript and moved up the figure 10 in the revised manuscript.
3. Statistics should be reported fully (e.g., even t-tests need associated degrees of freedom).
Response: thank you for your advice. We have provided full statistics in the revised manuscript (in the results section and the figure legends).
4. The paper is written in a way that suggests very little proof-reading was conducted. Above standard typos, which are usually not a huge issue to me, there are misplaced words, extraneous or missing words in some sentences, and several places just say “Error! Reference source not found!” in bolded font. Rather than walk through each of these individually, I would suggest a thorough examination of the manuscript before resubmitting (including figure captions, where many of these issues arise as well).
Response: thank you for your advice. We have thoroughly checked and edited for
language and form in the revised manuscript.
Minor comments:
5. Do the authors mean ‘vicious cycle’ rather than ‘vicious circle’? I would also suggest removing this term from the figures and instead just saying this triggers a cycle, rather than a vicious cycle.
Response: thank you for your advice. We have replaced “vicious cycle” with “vicious
circle” in the revised manuscript.
6. I would give a more thorough, albeit still brief, introduction to the role of epigenetics in AD symptomology, rather than just saying that changes are observed.
Response: thank you for your advice. The common epigenetics changes in AD patients contains abnormal DNA methylation, chromatin remodeling and histone modifications. Many clinical researches have suggested that DNA methylation levels of some genes could be potential biomarkers in AD. A range of studies indicates that histone modifications play a vital role in the development of AD. Especially, HDACs have significant influences on memory formation and cognition. We have added above statement to the introduction in the revised manuscript.
Round 2
Reviewer 1 Report
Authors responded to all reviewers' requests.
Author Response
Thank you for your comment.
Reviewer 2 Report
I still think there are issues with reporting of statistics and I still believe vicious circle should be corrected to cycle (see my previous review).
Author Response
Response to reviewer 2:
I still think there are issues with reporting of statistics and I still believe vicious circle should be corrected to cycle (see my previous review).
Response: thank you for your advice. We have checked all the results involving statistics and added the missing degrees of freedom in the revised manuscript. Furthermore, we added a section of statistical analyses which fully reported the statistical analyses in the methods part of the revised manuscript.
In the revised manuscript, we have removed the term of vicious circle. It is replaced by “rigger a cycle” or “form a cycle”.